# Simulation Comparisons of Particulate Emissions from Fires under Marginal and Critical Conditions

**Alexander J. Josephson \*** , **Daniel Castaño, Marlin J. Holmes** and **Rodman R. Linn**

Earth and Environmental Science Division, Los Alamos National Laboratory, Los Alamos, NM 88003, USA; dcastano@lanl.gov (D.C.); mjholmes@lanl.gov (M.J.H.); rrl@lanl.gov (R.R.L.)
\* Correspondence: alexanderj@lanl.gov

**Abstract:** Using a particulate emissions model developed for FIRETEC, we explore differences in particle emission profiles between high-intensity fires under critical conditions and low-intensity fires under marginal conditions. Simulations were performed in a chaparral shrubland and a coniferous pine forest representative of the southeast United States. In each case, simulations were carried out under marginal and critical fire conditions. Marginal fire conditions include high moisture levels and low winds, often desired for prescribed fires as these conditions produce a low-intensity burn with slower spread rates. Critical fire conditions include low moisture levels and high winds, which easily lead to uncontrollable wildfires which produce a high-intensity burn with faster spread rates. These simulations' resultant particle emission profiles show critical fire conditions generate larger particle emission factors, higher total mass emissions, and a higher lofting potential of particles into the atmosphere when compared against marginal fire conditions but similar particle size distrubtions. In addition, a sensitivity analysis of the emissions model was performed to evaluate key parameters which govern particle emission factor and particle size.

**Keywords:** CFD Simulation; Particulate Emissions; Soot Formation

---

## 1. Introduction

Smoke emissions from fires are governed by both processes in the source regime, which is the generation and immediate emission of species local to the flames, and in the plume regime, which is the subsequent evolution and transport of emissions. In recent decades, there has been an extensive research effort focused on the smoke processes occurring in the plume regime [1–5], but understanding of smoke processes in the source regime is still rudimentary at best. To quantify the response of emissions profiles to varying fire conditions, understanding smoke generation processes in the source regime becomes increasingly important.

To date, perhaps the most helpful work done with regard to smoke processes in the source regime comes from the development of a continually expanding emission factor inventory [6–8]. An emission factor is the mass of a particular emitted species over the total mass of consumed fuel. The inventory, which has been expanded with an extensive effort in the last few decades, contains experimental and field measurements of emission factors classified in a variety of ways. These classifications may come in the form of: the source fuel (i.e., biomass species [9]), ecosystem (i.e., grassland vs conifer forest vs boreal forest [10]), environment (i.e., laboratory vs open-field vs closed-forest [6]), or landscape (i.e., cityscape vs wildlands [11]), just to name a few. Unfortunately, inventories can differ greatly in their predictions and can be difficult to effectively use. In part, this comes from the extensive complications in measuring speciation of smoke effectively [12] and the variation in techniques or methods, used to obtain species identities and concentrations [6]. In addition, it can be difficult to inventory a fire by its characteristics: is it more important that a particular fire was predominately

ponderosa pine trees, or that it was a sparse forest, or that it was an uncontrolled wildfire, or that measurements were taken in the morning chill?

With regard to emissions' response to fire dynamics, Hoseini et al. [9] noted a general trend of higher particle emission factors from a heading fireline than from a backing fireline, this is generally extrapolated to mean that higher intensity fires, such as an uncontrolled wildfire, tend to have higher emission factors than low intensity fires, such as a prescribed fire. This trend was observed in field studies by Urbanski [7] who saw a 32% increase in particulate emission factor coming from wildfires in the northwest United States conifer forests when compared against prescribed fires in the same ecosystem. Neither Hoseini et al., nor Urbanski, provided an explanation for the trends they saw but both studies showed a correlation between higher fire intensity and higher particle emission factors. Combine this higher emission factor with a greater rate of fuel consumption as often observed in high-intensity fires [13] and one would expect to see higher concentrations of particles emitted to the atmosphere from high-intensity wildfires than from low-intensity prescribed fires.

In this study, we perform a series of fire simulations with a soot formation and emission model [14] to further explore trends in particulate emission profiles from fires under critical conditions and fires under marginal conditions. For the purposes of this study, marginal conditions lead to slow fire propogation with low intensity and are often sought by fire managers when prescribing fires. Critical conditions lead to fast fire propogation with high intensity and often set the stage for uncontrolled wildfires. The emission model was designed to predict the state of particles when they are emitted from the flames and does not include the post-emission evolution of particles. We also use the developed model to explore some of the causation for differences between these profiles.

## 2. Methods

Exploration of particulate emission profile trends were accomplished by performing four fire simulations:

1.　chaparral field with marginal conditions
2.　chaparral field with critical conditions
3.　conifer forest with marginal conditions
4.　conifer forest with critical conditions.

### 2.1. Simulations

Simulations were executed using the HIGRAD/FIRETEC computational fluid dynamics software. HIGRAD is an atmospheric hydrodynamics model which simulates airflow and its adjustments to terrain and fuel. FIRETEC, a wildfire behavior model, simulates combustion, heat transfer, aerodynamic drag, and turbulence at the fuel level. The coupling of the two captures interactions between fire and the atmosphere to predict wildfire behavior across complex fuel beds and terrain. HIGRAD/FIRETEC performs simulations by continually resolving equations based on conservation of mass, momentum, species, and energy [15].

Implemented within the HIGRAD/FIRETEC software is a zonal model for predicting soot formation and emission [14]. This model predicts the emission factor of 'soot particles' based on soot formation mechanisms established in literature. These mechanisms are imposed on a series of zones which constitute the structure of a flame: inception zone, heating zone, reaction zone, and quenched zone. The amalgamation of soot formation mechanisms with the zonal model for predicting flame structure results in sensitivies of this model to fuel composition, rate of fuel consumption, oxygen depletion, turbulent dissipation, kinematic viscosity, and wind velocities.

In these simulations, we assume the source term of $PM_{2.5}$, particulate matter with a diameter less than 2.5 μm, to come primarily from emitted soot particles. We define soot as particles formed within the flame by established mechanisms of the soot formation phenomena [16]. These soot particles should not be confused with black carbon, which are particles optically measured to be 'black'. In

the past, soot particles measured from diesel engines were observed to be entirely (or nearly so) optically-black and it is often that soot particles emitted from all sources are therefore black. In the simulated fires, we use a biomass as a fuel source, which chemically varies greatly from diesel fuel. These differences in chemistry lead to differences in the chemistry of emitted particles, which in turn affects optical properties. Thus in these simulations the predicted emitted particles are all $PM_{2.5}$ but not necessarily all black carbon.

The domain of each simulation was 800 m by 400 m discretized into 2 m by 2 m cells with a logrithmic vertical stretching component. Flat ground and a surface ignition in the form of a 8 m by 200 m fireline initiated 40 m into the domain was used for each simulation. Winds are roughly logarithmic with velocities set to a prescribed value 10 m above foliage levels. Before the fire simulation was carried out, winds were allowed to pass cyclically through the domain for a long-period of time to establish the turbulent profile and boundary layer in a 'wind-run'. During the fire simulations, boundaries of the simulated domain were continually updated with the boundaries of this wind-run to maintain the turbulent profile and boundary layer in the incoming air. Unlike the wind-run, during fire simulations the cyclic boundary conditions are no longer enforced so as to not recycle hot gases and smoke.

Marginal fire conditions were set with low-winds, 2.5 m/s, and high-moisture content, 125% of normal conditions which are elaborated on in the following sections. Critical fire conditions were set with high-winds, 12 m/s, and low-moisture content, 75% of normal conditions.

### 2.2. Chaparral

The first pair of simulations were of chaparral fields representative of the chaparral biome common throughout California. A chaparral biome consists of fields predominately occupied by shrubs of the chamise and ceanothus species. This fuel naturally occurs at lower moisture contents (40–90% [17]) and at higher densities (50–95% shrub cover [18]) than conventional forests. The combination of low moisture contents and high fuel densities lead to increased fire danger in these ecosystems and thus a ecosystem of interest for this work.

In this study, simulated fuel beds consisted of a evenly distributed field of shrubs with a 60% shrub cover, and a light loading of grass and litter in the voids between shrubs. Individual shrubs contained a crown diameter of 2.5 m and heights varied between 1–2 m. 'Normal' moisture content of shrubs was taken to be 65%, translating to a marginal moisture content of 81.25% and critical moisture content was 48.75%. Normal moisture content for ground fuels was 5%, implying 6.25% for the marginal conditions and 3.75% for the critical conditions.

### 2.3. Southeast US Conifer Forest

For the second pair of simulations, a representative plot of a conifer forest from the Eglin Air Force Base in Florida was used. This is an area of land commonly subjected to fire prescription and thus another area of interest for this work. Trees in this forest are predominately longleaf pines with frequent turkey oaks and occasional persimmons. Tree dimensions varied: longleaf pines heights were 10–20 m with crown diameters 2–10 m, turkey oak heights were 6–12 meters with crown diameters 2–5 m, and persimmons heights were approximately 8.5 m with crown diameters around 6 m. Normal moisture contents for each of these species were 133%, 200%, and 170% for the longleaf pine, turkey oak, and persimmons respectively. On the ground level, grass and litter was spread based on the proximity to trees and normal moisture content was 5%.

## 3. Results

Figure 1 is a visualization of the described simulations, with the bulk mass of fuel shown in green. Cell-averaged gas temperatures above 310 K, are shown in red to represent where active burning is occuring. The bulk mass density of emitted smoke particles is visualized on a logarithmic grayscale where black represents a bulk mass density of $1 \times 10^{-5}$ kg/m$^3$ or greater and white represents a bulk

mass density of $1 \times 10^{-8}$ kg/m$^3$. Each simulation was visualized at a different time step to show the resulting plume after the fireline had arrived at approximately the same location. For the conifer forest, this corresponded to the location where the head of the fire reached approximately 600 m and for the chaparral scrubland was where the head of the fire reached approximately 350 m.

The figure shows some interesting characteristics about each fire. First note the depth of intense burning, or fireline depth (the area which is colored red), in the Chapparal cases, especially Image (b). This wider fireline depth is due to the fast rate of spread and the fuels' longer burn-out time in the chaparral ecosystem and impacts the lofting of hot gases and thus transport of the emissions. The changes in plume characteristics associated with wider flame depth is known to significantly contribute to the global environmental impacts of emissions through the formation of pyrocumulonimbus clouds [19]. Even Image (a) which is chaparral fire under low intensity conditions has a fairly deep flame when compared to the conifer forest and may be something to be considered upon ignition depending on management goals in the case of prescribed burns.

(**a**)         (**b**)

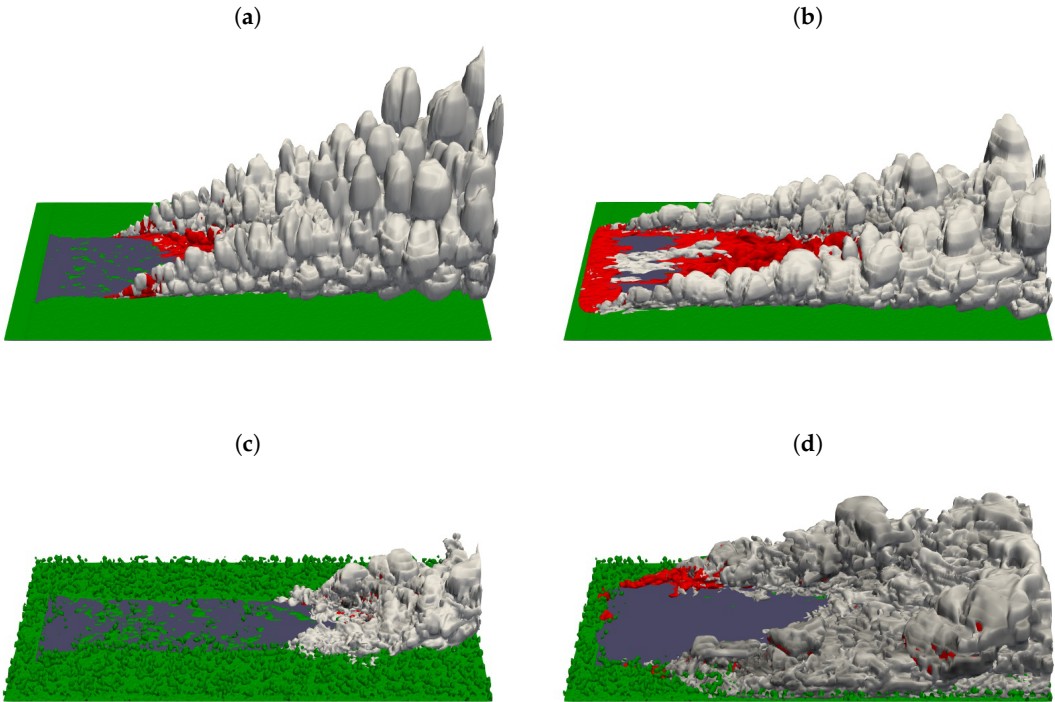

(**c**)         (**d**)

**Figure 1.** FIRETEC simulations results for the marginal fire and critical fire conditions. Visualized is the bulk mass density (kg/m$^3$) on a grayscale from $1 \times 10^{-6}$ (white) to $1 \times 10^{-2}$ (black) kg/m$^3$. Image (**a**) is the marginal fire condition chaparral case at 940 s, (**b**) is the critical fire condition chaparral case at 260 s, (**c**) is the marginal fire condition conifer forest case at 1750 s, (**d**) is the critical fire condition conifer forest case at 450 s.

Note that the conifer forest case under marginal conditions (Image (c)) has a significant amount of fuel leftover within the fire-scar. This is because under these conditions this fire rarely crowned but mostly remained on the ground with the occasional torching. In contrast, under critical conditions this fire easily crowned and completely consumed all fuels on both the ground and in the canopy as seen in Image (d).

At first glance it would appear that the marginal fire lofts particles higher than the critical conditions but this is just a characteristic of the greater winds sweeping the smoke cloud off the resolved domain before the full lofting height can be reached. Section 3.4 will more fully explore the lofting potential of each simulated fire.

### 3.1. Fireline Intensity

Byram's fireline intensity ($I_B$), is a measure of energy released per a unit of length of fire front $\left(\frac{kW}{m}\right)$ [20]. This metric effectively removes the impact of lateral extent of the fire and as such is often used to compare magnitude or scale, potential damage and the difficulty of controlling a given wildland fire. While fireline intensity is usually estimated from an observational measurement, for example flame height, which is then related to an empirical relation to determine intensity. With simulation data, we calculate this quantity directly:

$$I_B = \frac{\Delta \rho_f \Sigma V_{cell}}{\Delta t} \frac{H_f}{L_F} \tag{1}$$

where $\Delta \rho_f$ denotes the amount of fuel consumed in burning cells between two given time steps. $\Sigma V_{cell}$ denotes the sum of the volume of all burning cells, which is divided by the elapsed time between each time step $\Delta t$. $H_f$ represents the lower heating value of the fuel being consumed, while $L_F$ is the calculated fireline length.

As seen in Figure 2, differences in the magnitude of fireline intensity between marginal conditions and critical conditions are readily apparent. Critical conditions exhibit nearly an order of magnitude larger heat release, and therefore fireline intensity, driven by the higher wind speeds and lower fuel moisture content. Additionally, the length of time that data is available is much shorter for these simulations which is a consequence of their faster spread rates. These simulations "raced" across the domain when compared to marginal fire conditions.

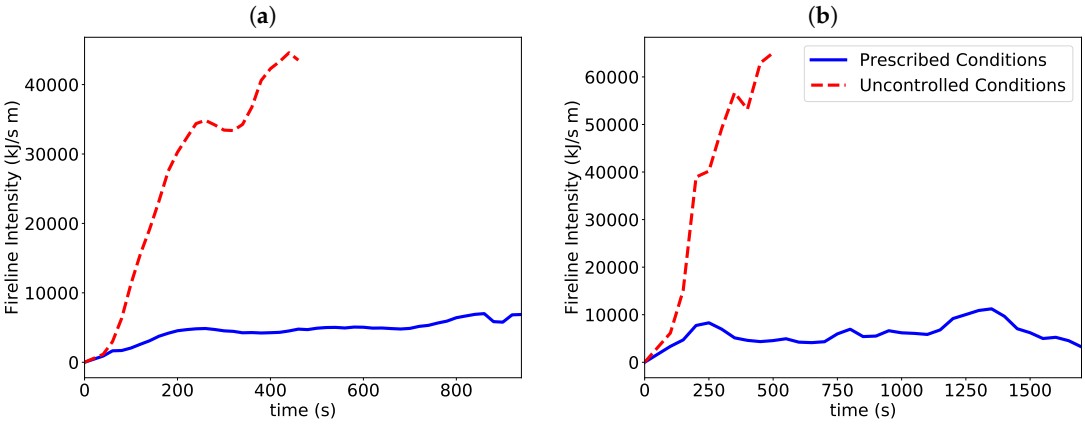

**Figure 2.** Fireline intensities over time from FIRETEC simulations results for the fires with marginal and critical conditions. (**a**) is a comparison for the chaparral ecosystem and (**b**) is the conifer forest. In each, the blue solid line represents the marginal fire conditions and the dashed red line represents the critical burn conditions.

The differences in intensity between critical and marginal conditions in Image (b) of Figure 2 is approximately fifty percent greater than the differences in Image (a). It should be noted that fuel arrangement may be partially responsible for these differences. The chaparral field, Image (a), contains only ground fuels as shrubs were continuous from ground to crown and were limited to 2 m in height; on the other hand, the conifer forest, Image (b), contains a combination of ground and crown fuels in distinct layers. The interplay of wind within and between these layers and the dynamic interplay between ground and crown fires may have worked in tandem to produce these differences in fireline intensities.

### 3.2. Emission Quantities

Table 1 summarizes the mass of particles produced by each type of fire. Quantities of emitted particles are directly proportional to the amount of fuel consumed by a fire and Lines 2 & 3 of Table 1

clearly show this correlation. However, perhaps more informative in these simulations is the difference in emission factor produced by each fire. For both simulated cases, the critical burn produced a higher emission factor than the marginal burn conditions. In the chaparral case, the critical burn produced an emission factor 2.3 times higher; whereas in the conifer forest case the emission factor for the critical burn was 3.4 times higher. These results may be more extreme, but are consistent in trend with observations made by Urbanski [7] who also observed a particulate emission factor increase in northwest conifer forests when moving from low-intensity prescribed fires to high-intensity wildfires.

**Table 1.** Summary of emitted particles from simulations.

|  | Chaparral | | Conifer Forest | |
| --- | --- | --- | --- | --- |
|  | Prescribed | Uncontrolled | Prescribed | Uncontrolled |
| Particulate Emission Factor | 1.59 | 3.67 | 5.96 | 20.14 |
| Total Mass Consumed (tonne) | 30.4 | 35.0 | 61.8 | 104.7 |
| Total Mass of Emitted Particles (kg) | 48 | 128 | 368 | 2,109 |

This increase in emission factor does not come as a surprise. Hosseini et al. [9] noted a higher emission factor from a heading fireline than from a backing fireline, this is generally extrapolated to mean that higher-intensity fires tend to have higher emission factors. The exact reasoning for this correlation is unknown, but we suspect it is due to the increased turbulence associated with high-intensity fires. Increased turbulence will shorten the reaction layer of a flame, the interface between fuel-rich and fuel-lean regions. It is in the reaction layer of a flame that particles are either partially or fully oxidized and thus by shortening the thickness of this layer, through increased turbulence, the fire will effectively emit more particles. In the critical fire conditions, higher wind velocities and lower moisture contents naturally create more intense fires, as seen in Section 3.1, and thus increase particle emission factor. However, the correlation between fireline intensity and particulate emission factors is non-linear; one more complicated by the interplay of several factors. In Section 3.5 we will explore some of these factors and their effect on particulate emissions.

In Figure 1, Image (d) has a much darker and larger smoke plume than Image (c) reflecting higher concentrations of particles produced in a larger area shown quantitatively in Table 1. However, when comparing Image (a) to Image (b) it appears that the smoke plume of Image (a) is larger and darker. We believe this is due primarily to the difference in wind velocities; in the critical burns, wind velocities were much higher than the marginal burns. This faster lateral wind does two things: first, it blows much of the smoke plume flatter and largerly out of the resolved domain thus even though Image (b) produced more particles than Image (a), shown in Table 1, most of the resulting plume is not depicted. Second, the incoming wind dilutes particle concentrations, in the case of the conifer forest there was so much particulate mass produced that we didn't see this dilution much but in the chaparrel case the burn is producing an order of magnitude less particulate mass. Due to the lesser emitted mass, wind dilution becomes more effective and thus the plume of Image (b) is less concentrated than that of Image (a).

### 3.3. Particle-Size Distributions

Figure 3 compares predicted particle-size distributions (PSDs) of emitted particles from simulated fires. Image (a) of the figure shows the two fires in the chaparral ecosystem while Image (b) shows the conifer forest ecosystem. The shown PSDs are emitted particles at the source, implying particles as they are emitted from the flame without any continued particle evolution within the smoke plume. When considering the potential human health effects of fires, this figure shows that under all tested conditions nearly all emitted particles would qualify as either $PM_{2.5}$ or ultrafine particles, diameter smaller than 1 μm. Both have been identified for their adverse health effects [21].

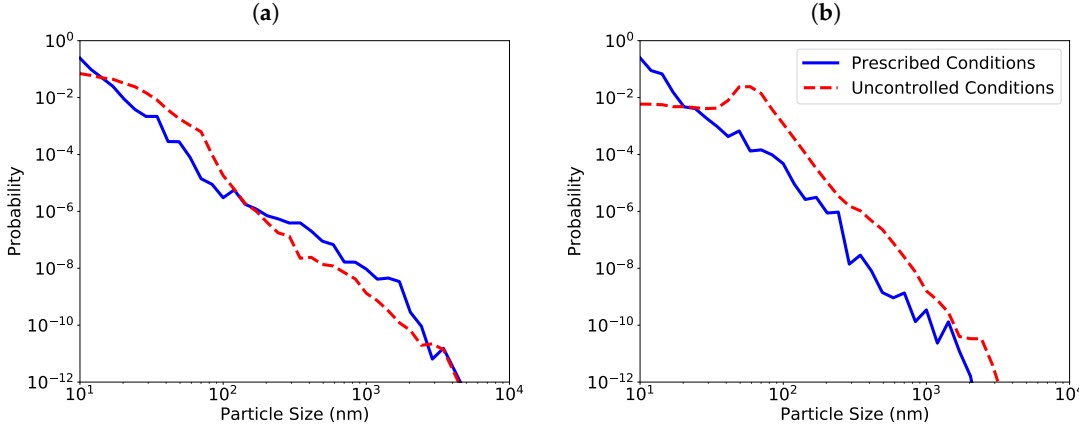

**Figure 3.** Particle-size distributions from the FIRETEC simulations results for the marginal and critical fires. (**a**) is a comparison for the chaparral ecosystem and (**b**) is the conifer forest. In each, the blue solid line represents the marginal burn conditions and the dashed red line represents the critical burn conditions.

From Image (a), it is shown that the overall PSD does not change significantly moving from marginal to critical fire conditions. The functions are not identical, and it seems that the marginal fire conditions may shift the distribution slightly towards larger particles, probably due to reduced mixing from weaker winds and less intense turbulence. This reduced mixing allows for particles to have longer residence times within the flame and thus grow larger through particle-particle agglomeration. However, we note here again that the shift is small and perhaps statistically negligible to draw any strong correlations from.

Whereas Image (a) showed almost no perceivable differences in PSDs, Image (b) shows a distinct, albeit small, shift of distributions towards larger particles when we move to critical fire conditions. This shift most likely occurs because under marginal conditions the ignited fire stayed at the ground-level whereas under the critical conditions the fire quickly crowned into the canopy. These canopy fires may produce larger particle for a few reasons. (1) Canopy fires tend to be more intense than ground fires and consume significantly larger quantities of oxygen which diminishes the rate of particle oxidation within the reactive layer of a fire. (2) Canopy fires have higher rates of mixing and more turbulence because there is air underneath the fire as well as above, this increased mixing and tubulence can significantly diminish the thickness of the reaction zone and thus reduce the total amount of particle oxidation. (3) Understory winds may stretch out the total length of a flame allowing longer particle residence times within the flame and thus more time for particle-particle agglomeration before particles enter the reactive layer. Despite multiple driving factors towards larger particles, even with the shift of the total PSD in Image (b) all particles still qualified as $PM_{2.5}$.

*3.4. Lofting Potential*

Wildland fires produce significant heat release leading to the generation of strong buoyant plumes which readily loft particles thousands of meters into the atmosphere. The altitude reached by these emissions can have significant consequences on smoke dispersion, air quality, and atmospheric chemistry. As fire convective heat release, $Q_c$, directly leads to the buoyant flux, $F_o$, of a given plume it can be readily shown that fireline intensity ($I_B$) and fireline length ($L_F$) can be leveraged to estimate the maximum height a given parcel of fluid and particulate may reach in the atmosphere as a function of time. This theoretical maximum plume rise, referred to as the 'lofting potential' or $Z_P$ in this work, is computed as follows [22]:

$$Z_P = 3.79 F_o^{1/4} G^{-3/8} \tag{2}$$

$$F_o = \frac{g Q_c}{T_o \rho_o c_p} \tag{3}$$

$$G = \frac{g}{T_o}\frac{dT_a}{dz} \tag{4}$$

$$Q_c = I_B L_F \tag{5}$$

where $G$ is the density stratification parameter determined by Morton et al. [23], $g$ is the acceleration due to gravity, $T_o$ and $\rho_o$ are the ambient temperature and density at the plume source respectively. $c_p$ is the coefficient of specific heat of ambient air at constant pressure, assumed constant at all heights. $\frac{dT_a}{dz}$ is the environmental lapse rate of temperature at sea level international standard atmosphere (ISA) which is defined by the International Civilian Aviation Association (approximately $6.5\frac{K}{km}$).

Figure 4 depicts the lofting potential of all simulated cases. As one would expect, larger fireline intensity leads to higher heat releases, and therefore larger buoyant fluxes which can carry particles higher aloft. It should be noted that when compared to Figure 2 the character of the curves as well as the relative magnitudes between the marginal and critical cases are not as large. This can be readily explained by the expression stated earlier defining lofting potential. When all other quantities are held constant, an increase in heat release leads to an increase in buoyant flux scaled by the one fourth power. As such, an order of magnitude increase in heat release rate produces nearly double the lofting potential. While the values of Figure 2 may seem high, a plume reaching heights of two thousand meters is well within bounds of the troposphere ( 7 to 20 km) and still much lower than the heights attained by plumes produced by "mega fires", which can easily reach stratospheric heights [24]. Additionally, these values were calculated neglecting any influence of the inversion layer which would certainly impact the maximum altitude these particles reach but changes according to the time of day as well geographic location [25].

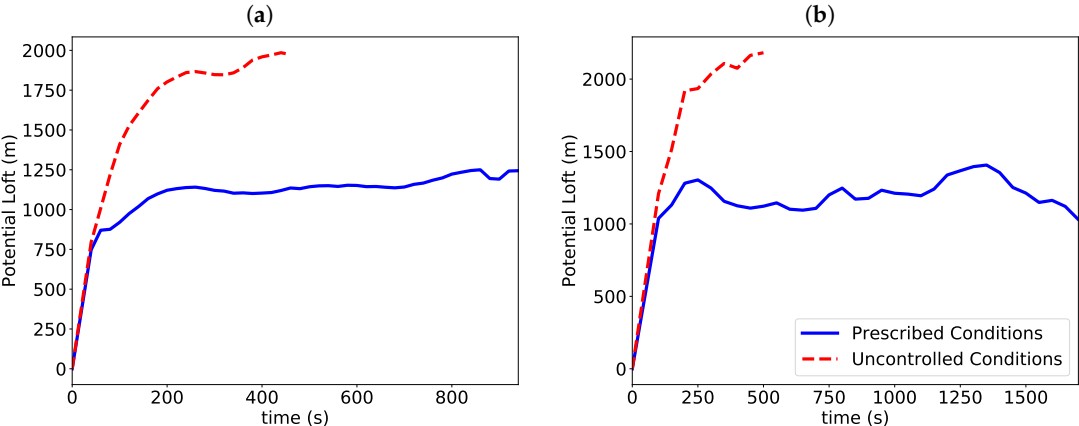

**Figure 4.** Lofting potential of particles over time from the FIRETEC simulations results for the fires under marginal and critical conditions. (**a**) is a comparison for the chaparral ecosystem and (**b**) is the conifer forest. In each, the blue solid line represents the marginal burn conditions and the dashed red line represents the critical burn conditions.

### 3.5. Sensitivity of Model

While the previous sections have explored differences in emission profiles, this section will begin to explore the why behind these differences. Assuming the used model [14] adequately represents the process of formation and emission of particles, we performed a sensitivity analysis of the model to the different input conditions found within the fire simulations.

Within a simulation, there are six primary inputs, and several secondary, to the particle emission model which governs the output emission factor and average particle size. These six primary inputs include: extent of reaction (aka, rate consumption of fuel), gas $O_2$ mole fraction, turbulent dissipation, bulk gas density, bulk solid fuel density, and local gas velocities. To determine the range which each parameter varies we carried out a grass simulation similar to those described above and tracked the

minimum and maximum value of each parameter inputed to the emission model through the duration of the simulation. The results of this simulation are summerized in Table 2.

**Table 2.** Ranges of the emission model's input parameters during a high wind grass simulation.

|  | min | max |
|---|---|---|
| Extent of Reaction | $3.05 \times 10^{-11}$ | 0.426 |
| $O_2$ Mole Fraction | 0.0145 | 0.210 |
| Turbulent Dissipation ($m^2/m^3$) | 0.0 | 18.61 |
| Gas Density ($kg/m^3$) | 0.268 | 1.159 |
| Bulk Fuel Density ($kg/m^3$) | 0.0142 | 0.464 |
| Velocity Magnitude ($m/s$) | 0.00340 | 25.69 |

The particle emission model was adapted to be executed independent of a fire simulation in order to test the sensitivity of this model to variations in input parameters. Each of the ranges in Table 2 were discretized into 30 individual values and the entire matrix of $30^6$ combinations of input discretizations were individually ran. Figure 5 shows a summary of these combinations where all combinations with the same input parameter are averaged together to give individual response curves to each primary input range.

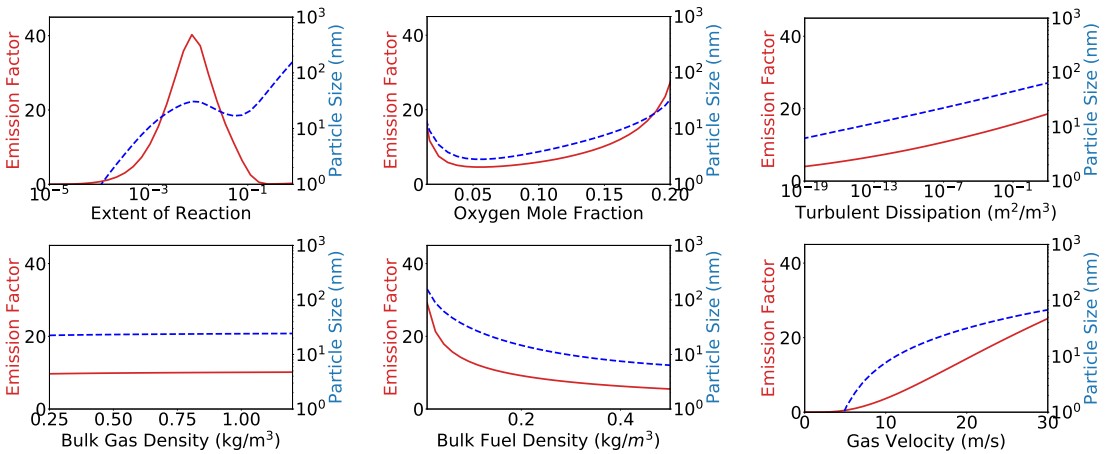

**Figure 5.** Response surfaces showing the sensitivity of predicted emission factors (red) and average particle size (blue) to the primary model inputs. Global 7-dimensional response surface is nominalized through averaging for each individual input.

Of the shown correlations, perhaps the most interesting come from the extent of reaction and oxygen mole fraction curves. The extent of reaction curve intially shows that the more fuel consumed the higher the emission factor and larger the particles. This is a direct correlation between more fuel providing more soot precursor molecules which leads to more soot particles emitted. However, at a point there is a turn where the emission factor decreases with increasing extent of reaction; this occurs because as we consume more and more fuel the fire becomes more spread out increasing the flame surface density and decreasing the overall flame length. This decreased flame length leads to less particle agglomeration within the flame and thus when particles enter the flame reaction layer they are more easily oxidized and consumed.

The oxygen mole fraction curve shows a general increase of emission factor and particle size as mole fraction increases. At first this seems conterintuitive because one would think that more oxygen means more oxidation, or particle consumption, and thus less emitted particles. However, one of the base assumptions of this model is that a flame goes from a fuel-rich region to atmospheric chemistry conditions. Thus oxygen mole fraction is not used to determine the chemistry of the flame, that is modeled in other ways, rather it is used to determine the overall size of the flame. Thus we

can interpret this plot to be more a reverse correlation between overall flame size and emission. In other words, as the overall size of the flame gets smaller the overall emission factor increases. This correlations has less to do with the overall size of the flame but more with the thickness of the reaction layer of the flame. In general, larger fires have thicker reaction layers which in turn increase the overall oxidation and consumption of particles.

The other correlations shown in Figure 5 tend to be a little more staight-forward. Turbulent dissipation and gas density are inversely proportional the reactive layer thickness, and a thinner reactive layer means less oxidation and more emitted particles. Gas density is only weakly influential; as a result, variations in gas density have a negligible effect on overall particle emissions. Bulk fuel density is directly proportional to reactive layer thickness leading to more oxidation and less emitted particles. Gas velocity influences many of the aspects of the model (residence time of particles in the flame and in the reactive layer, entrainment of air into the chemistry of the flame, etc.) and while some repress particle emission, other aspects more heavily promote the emission of particles with the increase of overall gas velocity.

The curves of Figure 5 beg the question– which of the input parameters becomes the most important to governing particle emission? To answer this question we compute the overall gradient response of each output to changes in input parameters. These gradient response values are shown in Table 3 listed from most important to least important parameter. This gradient response is computed by summing the gradients of the global response surface with respect to each individual primary input at each input permutation. This sum is then normalized by the range of input values to give a comparable number. In essence, the larger the gradient response value, the more influence perturbations in that variable will have on overall particle emission factor or particle size.

**Table 3.** Gradient response of each of the model's primary input parameters normalized across the ranges of those parameter given in Table 2.

|  | Emission Factor | Particle Size |
|---|---|---|
| $O_2$ Mole Fraction | 76.5 | 471.6 |
| Extent of Reaction | 80.7 | 183.8 |
| Bulk Fuel Density ($kg/m^3$) | 23.7 | 152.4 |
| Velocity Magnitude ($m/s$) | 25.1 | 67.6 |
| Turbulent Dissipation ($m^2/m^3$) | 14.9 | 58.4 |
| Gas Density ($kg/m^3$) | 0.5 | 1.8 |

We warn that these gradient responses consider the entire range of input parameters uniformly; however, in simulation input parameters are not uniform. For example, $O_2$ mole fraction is shown to be the most important factor in determining particle size when considering it's entire range, but in simulations we found that while $O_2$ mole fraction does vary across it's entire range, most of the time values will sit between 0.19 and 0.21 which is a much smaller range and thus less influential than the full range of 0.145 to 0.21. This distribution of parameters across their range will be fire dependent and thus $O_2$ mole fraction may be the most important factor in one fire whereas extent of reaction may be much more important in another. Although the distribution of parameters across their range may be fire dependent, the gradient response does not vary; thus while individual fire simulations may vary the importance of different input parameters it would be unlikely for that importance to drastically change. For example, it is unlikely that gas density would ever be important than extent of reaction for computing emission factor.

While the curve averaging and gradient response numbers allow us to see the sensitivity of the model to changes in individual parameters, we lose the coupling of inputs that occur. In addition, and perhaps more importantly, with this method we lose simulation coupling of inputs such as the strong correlation between turbulent dissipation and gas velocity. Nonetheless this excercise is helpful in understanding the relative importance of different inputs to forming and emitting particles from fire.

## 4. Discussion

The results of Section 3 indicate considerable advantages to prescribed burning versus uncontrolled wildfires from an emissions perspective. The higher intensity fires of uncontrolled wildfires lead to significantly higher rates of fuel consumption translating directly to higher quantities of particles emitted to the environment, but even should there be comparative quantities of fuel consumption, the lower emission factor of low-intensity prescribed burns still implies lower quantities of emitted particles. In addition to lower quantities, our simulations also indicate a higher lofting potential of particles into the atmosphere for high-intensity uncontrolled wildfires. These higher lofting potentials will translate directly to a larger area of dispersion for these particles implying a larger area of impact for any given fire. From an emissions perspective, if there is a disadvantage of a prescribed fire it is that emitted particles from ground fires will tend to be slightly smaller than particles emitted from canopy fires which frequently occur in uncontrolled wildfires. However, while there is a detectable difference in size, all predicted particles were still under the $PM_{2.5}$ threshold. It is the authors' belief that any health benefits of slightly larger particles are far outweighed by the sheer quantity and lofting height differences between low-intensity prescribed fires and high-intensity uncontrolled wildfires.

In this work, we made the base assumption that fire managers would prescribe low-intensity fires for use as a fuel management tool and that high-intensity fires lead to uncontrolled wildfires. However, this may not always be the case, as there are scenarios when fire managers may apply particular fire dispersion techniques to create small, but high-intensity fires to produce specifc outcomes. While our base assumption of desirability may differ in these cases, the results of this work are still applicable to such circumstances. Namely, higher-intensity fires will most likely produce higher quantities of particles that will be lofted higher than low-intensity equivalencies, but with similar PSDs.

### 4.1. Limitations to This Study

It is well known that post-emission particles will continue to evolve within the plume itself [26]. This continued evolution comes from particle-particle and particle-gas interactions as particles may continue to agglomerate [27], partially oxidize [28], or serve as a condensation site for other organic carbon-heavy gaseous species also emitted from the fire [26]. Our simulations do not reflect any of these evolutions but rather only reflect the state of particles when emitted from the fire itself. In addition, the developed model should not be compared directly to black carbon emissions. Measured black carbon emissions are often set by a comparison to soot particles emitted from diesel engines [29], although the developed model was designed to predict the emission of soot particles, we define soot particles as particles formed by the soot formation phenomena [16] with distinct mechanisms not by the particles' optical properties. The chemistry of soot particles emitted from biomass sources is known to be different than soot particles emitted from diesel engines [30] and how optical measurements may respond to the chemical differences is not entirely known.

FIRETEC and the developed particle emission model are designed to capture fire-spread accurately. Thus these models are predominatly focused on flaming combustion systems, as opposed to smoldering combustion. In consequence of these developments, we've chosen circumstances which we believe are primarily flaming combustion for this study and have limited the influence of smoldering combustion, which in other circumstances can play a much larger influence on emission profiles.

Lastly, we feel obligated to recognize that this study was performed completely with simulations using developed models in various stages of validation. While the results of these simulations reflect similar observations in laboratory and field observations noted throughout this article, exact quantities reported in this work should be used with caution. Rather the trends and observations made in this study should be applied.

*4.2. Conclusion*

In this study we have presented the results of four simulations. These simulations were of fires ignited in a chaparral and in a conifer forest ecosystem, each under marginal and critical fire conditions. In both ecosystems, the fire with critical conditions produced significantly higher quantities of particulate emissions with larger emission factors than the fires with marginal conditions. Thus showing that particle emission factors have a significant dependence on fire dynamics and not ecosystem alone. In addition, the critical condition fires had an order of magnitude higher fireline intensity and thus could potentially loft particles twice as high into the atmosphere. Given these results, where possible, it is more desirable, from an emissions prespective, to prescribed low-intensity burns rather than allow high-intensity fires to form at a later date.

**Author Contributions:** The following contributions can be attributed to each author. Conceptualization: A.J.J.; methodology: A.J.J. and M.J.H.; software: R.R.L.; investigation: A.J.J., D.C., and M.J.H.; data curation: A.J.J. and D.C.; writing—original draft preparation: A.J.J., D.C., and M.J.H.; writing—review and editing: M.J.H. and R.R.L.; visualization: A.J.J. and D.C.; project administration: R.R.L.; and funding aquisition: R.R.L.

**Funding:** Primary funding for this work was provided by the USDA Forest Service through the National Fire Plan and Rocky Mountain Research Station. Additional funding was provided by theAfrican American Partnership Program. Computational resources for the performed simulations were provided by the Los Alamos National Laboratory's Institutional Computing Program.

**Conflicts of Interest:** The authors declare no conflict of interest.

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
