# Peer review of "Simulation Comparisons of Particulate Emissions from Fires under Marginal and Critical Conditions"

_atmosphere, doi:10.3390/atmos10110704_

Round 1

Reviewer 1 Report

In this manuscript authors show a series of fire simulations with a soot formation or emission model to further explore trends in particulate emission profiles.

I have two comments for this manuscript as following.

The objective of the study in the Introduction section should explain that the study focuses on simulation of state of particle when it was emitted and not include the post-emission particles evolution.

Lines 305 ~ 312

Since the gradient response of each of the model’s primary input is fire dependable then does it mean all the parameters will have the same degree of influence?

Author Response

We'd first of all like to thank the reviewer for their consideration and quick feedback to the proposed article. Following is a response to the two concerns/questions put forward by the reviewer and the manuscript has been modified with highlighted changes and resubmitted. Concerns/questions are given in black, while the responses are in blue.

The objective of the study in the Introduction section should explain that the study focuses on simulation of state of particle when it was emitted and not include the post-emission particles evolution.

This has now been added to the introduction section, final paragraph.

Since the gradient response of each of the model’s primary input is fire dependable then does it mean all the parameters will have the same degree of influence?

Not at all, the input parameter distribution will vary fire-to-fire and that would change the order of importance for various input parameters, but the performed model sensitivity (gradient response calculations) give us a decent snapshot idea of the order of importance for input parameters outside of the context of a given simulation. So although we can contrive of fire scenarios where the order of parameter importance sees slight shuffling than the gradient response calculations, you’d be hard pressed to see more than a slight shuffling. In other words, gas density will probably always be the least important parameter for computing emission profile characteristics while extent of reaction of oxygen depletion will almost always be among the most important.

We’ve made additions to this section of the paper to hopefully make this more clear.

Reviewer 2 Report

The authors present a particle-generation model coupled to an existing CFD-based wildfire simulation to compare emissions from uncontrolled wildfires and prescribed fires. Their results reveal interesting differences between high and low intensity fires in two studied fuel types, chaparral and forested areas. The results are interesting and include not only comparisons of two different simulated cases, but also interesting physical insight based on a sensitivity study.

One question - is smoldering vs. flaming combustion explicitly modeled and does that play a large role in the emissions?

Starting on lines 169 – 170 – your discussion notes higher emissions factors for uncontrolled cases. It is not clear at this point in the paper what this is attributed to. Could some information be provided here in the paper? There is some discussion much later, but it is disjointed from where it is presented here. It might also be appropriate to point the reader to later discussion.

Smaller corrections noted:

Abstract – “out of a chaparral shrubland” – should that be “in chaparral”?

“under precarious fire conditions” – could you be more specific and not use precarious?

Line 42-43 – it may be worth mentioning this amount is expected to be beyond just the amount attributed to the effect of additional fuel consumption, is that correct?

Line 128-126 – wouldn’t this be due not only to the faster rate of spread and  a long enough burnout time to extend the depth of the burning region?

Figure 1 caption –“ (c) is the low-0intensity” – remove 0

Line 162 – add “of”

Line 174 -175 – Why did Hosseini et al find a higher emission factor? That would be interesting context in a sentence or two to add to the discussion.

Line 176 – “In the used uncontrolled” – delete used

Line 231 – Include Morton Taylor Turner Reference

Line 330 – put the 5 in PM2.5 subscript.

Line 334 – “ead” should be lead

Author Response

We'd like to thank the reviewer for taking the time and consideration for this article; particularly with such a fast turnaround time from the initial submission. We feel the review and the subsequent response to this review have made this work stronger and more complete. Following is a prepared response to the review with each concern or question repeated in black, and the response in blue. The article was altered in response to this and the other performed review and the changes were highlighted.

One question - is smoldering vs. flaming combustion explicitly modeled and does that play a large role in the emissions?

 You’ve hit upon perhaps the weakest aspect of this study. FIRETEC, in its current configuration, was not designed to capture smoldering combustion, but rather focuses on fire spread through flaming combustion. This does not mean that smoldering combustion is completely neglected but rather that our representation of it in these simulations contains the most uncertainty. As a result, we picked scenarios to model which we believe are dominated by flaming combustion. If we consider an ecosystem, such as the boreal forest, where smoldering combustion becomes increasingly important then this would play a larger role in the emissions profile and we’d not be as confident in our simulation results.

We’ve made additions to the ‘Limitations to this Study’ section of this paper to reflect your concern here.

Starting on lines 169 – 170 – your discussion notes higher emissions factors for uncontrolled cases. It is not clear at this point in the paper what this is attributed to. Could some information be provided here in the paper? There is some discussion much later, but it is disjointed from where it is presented here. It might also be appropriate to point the reader to later discussion.

 We’ve tried to add a few lines here connecting the higher emission factor to higher turbulence and succinctly explain the cause of this correlation. However, as you’ve noted the model sensitivity section covers the causation in much more detail so we’ve also add a pointer for the reader as recommended.

Smaller corrections noted:

Abstract – “out of a chaparral shrubland” – should that be “in chaparral”?

Fixed

“under precarious fire conditions” – could you be more specific and not use precarious?

We’ve changed this to ‘critical’ conditions and tried to make changes throughout the paper (including the title) to reflect this change. Now ‘prescribed’ conditions are noted as marginal conditions, and ‘wild’ conditions are noted as critical conditions.

Line 42-43 – it may be worth mentioning this amount is expected to be beyond just the amount attributed to the effect of additional fuel consumption, is that correct?

We haven’t made any changes here because we feel the stated higher emission factor implicitly states this.

Line 128-126 – wouldn’t this be due not only to the faster rate of spread and  a long enough burnout time to extend the depth of the burning region?

Correct, and changed in the paper

Figure 1 caption –“ (c) is the low-0intensity” – remove 0

Fixed

Line 162 – add “of”

Fixed

Line 174 -175 – Why did Hosseini et al find a higher emission factor? That would be interesting context in a sentence or two to add to the discussion.

Hosseini never states the why of the higher emission factor, so we can’t state his/her reasoning but we’ve tried to add a few lines of our own conclusions as stated above.

Line 176 – “In the used uncontrolled” – delete used

Fixed

Line 231 – Include Morton Taylor Turner Reference

Fixed

Line 330 – put the 5 in PM2.5 subscript.

Fixed

Line 334 – “ead” should be lead

Fixed

Round 2

Reviewer 1 Report

I have checked the revised manuscript. The authors have answered my question and concern.

I recommend this revised manuscript to be published in the Atmosphere.

Best wishes.